# A novel approach of overtaking maneuvering using modified RG method

Usman Ghumman[1]*, Hamid Jabbar[2], Mohsin Islam Tiwana[1,2], Ihsan Ullah Khalil[3], Faraz Kunwar[1]

1 Department of Mechatronics Engineering, NUST College of E&ME, Rawalpindi, Pakistan, 2 Robot Design and Development Lab (RDDL), National Centre of Robotics and Automation (NCRA), NUST College of E&ME, Rawalpindi, Pakistan, 3 Department of Electrical Engineering, NUST College of E&ME, Rawalpindi, Pakistan

* usman.ghumman@ceme.nust.edu.pk

**Data Availability Statement:** All relevant data are within the paper and its Supporting Information files. Data sets of all experiments, data sets of all scenarios, codes of simlations along with videos are uploaded.

## Abstract

Intelligent and safe overtaking maneuvering is always a challenging task for autonomous vehicles. This paper proposes and experimentally implements a novel approach of overtaking maneuvering using modified form of Rendezvous Guidance (RG) algorithm for trajectory planning and obstacle avoidance, considering driver safety and comfort during autonomous overtaking. The simulations for all possible scenarios are conducted to ensure the effectiveness of proposed modified RG algorithm. These scenarios involved presence and absence of obstacle vehicle in overtaking lane alongside leading vehicle in driving lane. In addition, the enhanced performance of modified RG algorithm is established over conventional RG algorithm by comparative analysis. The results indicate that overtaking maneuvering period could be decreased by 10% using a modified RG algorithm and vehicle will cover less distance to complete overtaking. The efficacy of proposed method is justified by performing experiments using mobile robots. The experimental results and simulation results of modified RG algorithm are compared, and their plots are almost identical.

## Introduction

The sales of Autonomous vehicles have been a center of attention for researchers nowadays as it enables vehicles to perform regular driving tasks automatically. Automatic driving has numerous advancements within the domain of lane-keeping, distance maintenance, cruise control, and lane departure, etc. Such improvements were vital in ensuring safety and comfort but unfortunately, several challenges are still associated with autonomous vehicles [1–4]. The most significant amongst them is decision making. Overtaking is one of the arduous tasks that come under the umbrella of decision making. It involves lateral and longitudinal motion to avoid collisions and includes various other tasks like lane-changing, lane-keeping, and returning to driving lane [5].

Previously proposed approaches were quite promising in terms of handling moving obstacle avoidance problems. Incremental search algorithm and sampling-based trajectory planning method like RRT is proposed for ensuring safe trajectories while overtaking [6, 7]. The Model

**Funding:** The author(s) received no specific funding for this work.

**Competing interests:** The authors have declared that no competing interests exist.

**Abbreviations: Nomenclature**, **Description**; RG, Rendezvous Guidance; MPC, Model Predictive Control; '$M$', Marker Target; '$L$', Leading Vehicle; '$C$', Chasing Vehicle; '$OB$', Blocking Vehicle; RL, Rendezvous Line; VL, Velocity Line; $\mathbf{v}_c$, Velocity of '$C$'; $\mathbf{v}_b$, Velocity of '$L$'; $\mathbf{v}_b$, Velocity of Shadow Target; LOS, Line-of-Sight; $\lambda$, LOS Angle with Fixed Reference (X-Axis); $h$, Distance between Chasing Vehicle and Shadow Target in a Lateral Direction; '$r$', Length of LOS; '$\mathbf{r}$', Relative Velocity; $t_n$, Remaining Time-to-Intercept from Current Instant; $t_{max}^{cr}$, Maximum Closing Velocity for Fast Asymptotic Interception; $V_{max}^{rel}$, Final Value of Allowable Closing Velocity Component; RS, Rendezvous Set; $\Delta t$, Time Interval; $a_{Ymax}$, Maximum Lateral Acceleration; h, Width of Lane; $\vartheta$, Maximum Turning Angle of Chasing Vehicle; $\delta$, Current Heading Angle of Chasing Vehicle; FVR, Feasible Velocity Region; $\mathbf{v}_{RG}$, Desired Velocity of Chasing Vehicle in Upcoming Instant by Modified RG Method; $\mathbf{a}_{RG}$, Desired Acceleration of Chasing Vehicle in Upcoming Instant by Modified RG Method; NFVR, New Feasible Velocity Region; $l$, Distance between Chasing Vehicle and Shadow Target in an Axial Direction; $A$, Acceleration of Chaser Vehicle; $t_{max}^{rend}$, Magnitude of Maximum Allowable Closing Velocity.

Predictive Control (MPC) has been utilized in [8] by having non-convex avoidance constraints in the optimization framework that limits the uniqueness and feasibility of an obtained solution. Authors in [9, 10] employed potential field method for trajectory generation by compromising on vehicle user comfort, emergency situations, and reaction of other (obstacle and vehicle to be overtaken) vehicles. Similarly, receding horizon control in [11] modeled surrounding vehicles as Markov Gaussian Processes and multi-policy decision making in [12] via geometric partition through Voronoi cells deals with lane changing problem but had same aforementioned drawbacks of user comfort, emergency situations, etc. In [13], authors proposed a practical approach for driver assistance during overtaking by analyzing the acceleration and lane of next vehicle from opposite direction. Radar and video camera sensors are used in sensors fusion concept. But it failed when multiple vehicles have same acceleration and moving in different lane. In [14], follow the gap algorithm was proposed for vehicle overtaking during tense situations i.e., multiple vehicles in same line and having same speed. Results were compared with sin-X algorithm [15] and were found comparatively better. But follow the gap algorithm is not efficient when next vehicle varies speed regularly or changes lane due to heavy traffic density. X-sin function uses mathematical function for analyzing the scenario for comfortable overtaking maneuvering. It calculates longitude and lateral position of the ego vehicle. Technology independent sensor (TIS) is implemented via MATLAB and Pre-Scan software's by [16], which scan geometries of the vehicles to assists driver in safe overtaking maneuvering. Acceleration and vehicle direction were not considered which reduced the accuracy. Hierarchical overtaking technique for driver assistance is proposed in [17] which uses acceleration and speed signals from the surrounding vehicles and uses them in clustering methods to achieve high probability density function, which results in predicting expected motion. Computed reference is then tracked using the Linear Parameter Varying (LPV) control design method to guarantee safe motion. This proposed technique is very complex and requires lot of computational power. It can handle high density traffic interactions. [18] proposed deep learning-based reinforcement framework method for generating vehicle motion model, but huge data set is required for training safe overtaking safe maneuvering strategy. In [19] trajectory optimization for safe overtaking is performed via behavior and trajectory planning algorithm. This technique minimizes intrusion onto the adjacent lane. The proposed technique is very slow in performance therefore it is not adopted for further implementation. But this technique is more accurate than all mentioned techniques. [20] proposed brix model for overtaking maneuvering of two tankers. Moment acting on the ship hulls and maneuvering motion were analyzed for safe overtaking. Special function was established for calculating minimum distance, but this technique is only valid for two vehicles. Behavior of 3010 vehicles were analyzed in [21] during overtaking in China. Nonparametric survival analysis was performed to model the overtaking time before conducting log-rank test. It was found that conventional vehicle is faster in overtaking than electrical vehicles, as well as men are also safer than women during overtaking. Dynamic trajectory planning algorithm is prosed in [22], in which overtaking maneuvering trajectory is divided into short time trajectories to manipulate safe overtaking. This method becomes complex for three lane road. In [23] overtaking maneuvering is taken as an optimization problem, pontryagain's minimum principle is used to reduce the fuel consumption by 18% during overtaking period. However, constraints of the said scenario do not fit for every vehicle and high-density traffic. In [24] optimal and shortest path selection for overtaking maneuver is done using optimization algorithms, but this is only valid when a single vehicle has many choices for overtaking and have very low traffic density on road.

Therefore, it is the need of an hour to devise a solution that shall address the safety and user comfort problems while overtaking.

The article proposes a motion planning method to enable pursuer vehicle to perform optimal decision for overtaking maneuver based upon Rendezvous Guidance (RG) technique. Leading vehicle changes speed and lane regularly, which is modeled in expected six scenarios. Initially, the scenario of an absence of any other vehicle in overtaking lane is considered. Once the distance between two vehicles gets 2s, RG algorithm creates imaginary target points '$M_1$', '$M_2$', and '$M_3$' to ensure successful overtaking of leading vehicle '$L$' by chaser vehicle '$C$'. The second scenario involves blocking vehicle '$OB$' in overtaking lane which inhibits '$C$' to overtake '$L$'. Therefore, proposed algorithm comes into effect and allows '$OB$' to overtake '$L$' first to create a room for '$C$' to overtake '$L$'. The third scenario involves cancellation of overtaking decision after '$C$' gets into overtaking lane. This situation arises when '$L$' starts accelerating and the velocity of '$L$' becomes greater than '$C$' and eventually, '$C$' needs to get back in driving lane. The final situation is complex in comparison with previous situations as it involves multiple vehicles '$L_1$' and '$L_2$' in driving lane and also the blocking vehicle '$OB$' in overtaking lane. In this scenario, RG algorithm will not allow '$C$' to cross both vehicles straight away as it waits for a particular distance between '$OB$' and '$L_1$'- '$L_2$' to ensure the '$C$' gets safely back into a driving lane after overtaking '$L_1$' and '$L_2$'.

The aforementioned scenarios were simulated on RG technique and its modified form. The movement in highway is highly predictable therefore, instead of using Rendezvous line (RL), velocity line (VL) is used for efficient lane changing and overtaking decision making. A comparative analysis has been performed for RG technique and its modified form via simulations. The results demonstrate that modified form of algorithm allows 10% less time for chasing vehicle to overtake a leading vehicle and thus less distance is covered by chasing vehicle while overtaking. The experimental setup for comparison of both approaches are also elaborated by a specification of hardware equipment involved. Two different experiments are performed for justification of simulated results of modified RG algorithm.

The major contribution of the article are as follows.

1. Devising a methodology based upon RG technique that not only performs well by treating vehicle in driving lane as an obstacle but also yields effective results in presence of blocking vehicles in overtaking lane alongside leading vehicles in driving lane.

2. A novel modified form of RG technique is proposed and its comparative analysis with conventional RG technique is also conducted to validate the effectiveness of the proposed approach.

3. Modified RG technique is also compared with conventional offline technique.

4. Four different scenarios are modeled, which caters almost maximum possible situations.

The uniqueness of manuscript is portrayed by an attempt to address the scenario of overtaking which involves multiple vehicles (driving and overtaking lane) through RG technique and proposed modified form of RG technique. The proposed technique yields better results as it allows 10% less time for overtaking. In addition, the decisions in these complex overtaking scenarios have always been made considering the safety and comfort of chasing vehicle user.

The manuscript comprises of six sections. The research scenario is introduced in first section while the second section elaborates the problem description and proposed approach to tackle the discussed problem. The overtaking scenarios and decision making are highlighted in third section. Section IV covers the simulation results for all scenarios using modified RG algorithm and comparative analysis of modified and conventional RG algorithm. The experimental results and simulation results are compared in fifth section to verify the applicability of proposed algorithm alongside the description of the experimental setup. Finally, the manuscript is concluded in sixth section.

## Problem formulation

As mentioned earlier, the primary objective of our research work is to devise a methodology which provides safety and comfort to vehicle users while overtaking. Therefore, it is deemed necessary to discuss user comfort in detail while overtaking as we will be incorporating parameters in our proposed mathematical model to ensure the comfort of vehicle owners. The ride comfort is evaluated upon acceleration and angular motion. The sudden motion causes intense comfort disturbance and high lateral acceleration will result in indirect comfort disturbance [25]. To ensure a comfortable ride, the value of lateral acceleration and axial acceleration shall not exceed 1.25 m/s2 and 5 m/s2 respectively [26, 27]. The time interval of 2s is considered an ideal value for a safe distance between two vehicles to ensure braking operation without collision [28–30]. The values of ride comfort and braking distance are incorporated as constraints in a mathematical model. The model is focused upon the development of safe and optimal trajectory for lane changing to ensure seamless overtaking process.

## Three and four phase overtaking maneuver

To understand the mathematical model in detail, the description of terminologies and potential scenarios involved is required. We begin with a simplest lane changing scenario in which we have labeled the chasing vehicle as '$C$' and leading vehicle with '$L$'. If '$C$' wants to overtake '$L$' then the velocity of chasing vehicle must be greater than a leading vehicle. The velocity of '$C$' is indicated by $v_c$ and velocity of '$L$' is given by $v_l$. It is assumed that there is no blocking vehicle '$OB$' in overtaking lane and thus the process of overtaking is reduced to three stages only. These three stages are overtaking or lane changing maneuver i.e. maneuver from driving lane to overtaking lane, travelling in overtaking lane, and ultimately returning to the driving lane.

If the blocking vehicle '$OB$' is present in overtaking lane, it will not allow '$C$' to overtake. Therefore, '$C$' has to adjust its velocity in a manner so that once overtaking lane is cleared by '$OB$', only then '$C$' proceeds to overtake '$L$'. In case of presence of a blocking vehicle in overtaking lane, the stages of overtaking maneuver are increased by 1 as fourth stage involves velocity adjustment. The whole scenario is depicted in Fig 1. Upper lane in Fig 1 is overtaking lane while lower lane is travelling lane. Same is the case in all scenarios figure.

**Rendezvous Guidance technique.** The optimal overtaking maneuver for chasing vehicle can be planned via Rendezvous Guidance (RG) technique. RG technique is originally introduced for spaceships rendezvous missions with space stations and asteroids [31, 32]. RG technique has delivered promising results for non-maneuvering targets interception which corresponds to vehicles moving in a straight path in our case. Therefore, this technique is adopted in our research article, in which six different scenarios are analyzed. However, there

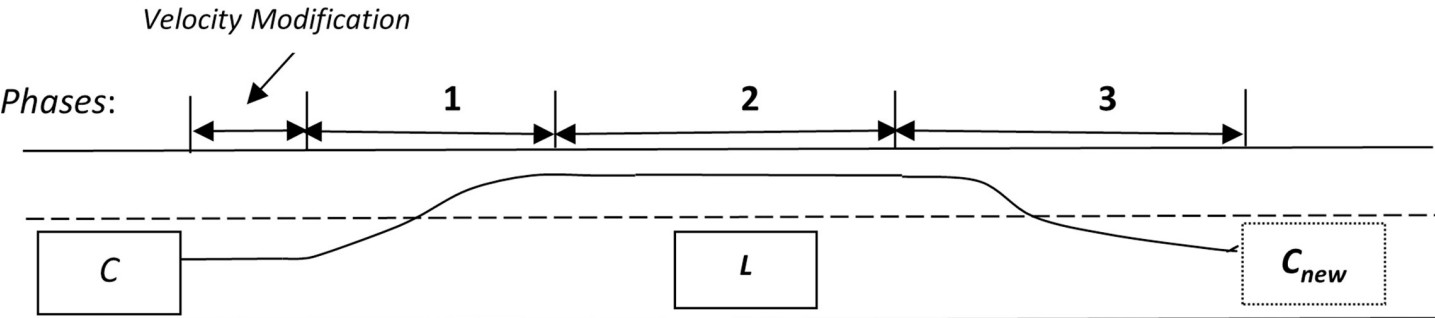

**Fig 1. Stages of overtaking in complex scenario.**

are two major limitations associated with RG technique. The first limitation is RG technique is designed for rendezvous missions with a particular target and it does not include overtaking the target. Secondly, numerous constraints are attached to chasing vehicle which are not associated with spaceships, so an amendment is required in technique to make it useful for optimal overtaking maneuver.

The first limitation is addressed by the introduction of a shadow target. Shadow target will guide 'C' through all the phases of overtaking to ensure user comfort and safety. The location of shadow target will be defined according to the location of vehicle which is to be overtaken. The second limitation is addressed by adopting RG technique proposed in [33–37] for robotic autonomous vehicle interception. The technique mentioned in [33–37] enabled us to gather relevant information of 'C', 'OB', and 'L' for generation of single acceleration command to be used for 'C'. The acceleration command would enable 'C' to overtake 'OB' and simultaneously avoiding 'L' in a time-optimal manner. The acceleration command is calculated via velocity-matching with shadow target considering constraints applied due to chasing vehicle dynamics and user comfort. The shadow target will also avoid vehicle collision in overtaking lane by avoiding obstacle. When RG is compared to other overtaking techniques, it results in higher accuracy, less complexity and can model maximum scenarios.

**Rendezvous Guidance law based trajectory.** The understanding of RG law-based trajectory requires discussion of relevant parameters in two-dimensional geometry to help readers grasp the functionality of RG technique. Consider two-dimensional geometry which involves 'C' and shadow target having a velocity of $v_c$ and $v_t$ respectively. The imaginary line connecting 'C' and shadow target is referred as Line-of-Sight (LOS). The angle created by LOS with fixed reference (x-axis) as given in Fig 2 is given by $\lambda$ which is calculated as given in (1).

$$\lambda = \tan^{-1}\frac{h}{l} \tag{1}$$

Where $h$ is the distance between chasing vehicle and shadow target in a lateral direction and $l$ is the distance between them in an axial direction. The length of LOS is defined as range '$r$'. As per the parallel navigation law, the LOS direction shall remain constant with respect to non-rotating frame while chaser approaches the target. Therefore, the relative velocity

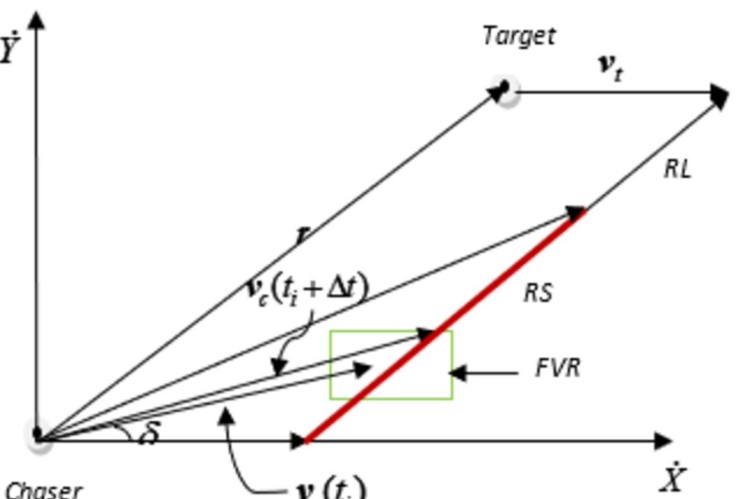

**Fig 2. Construction of Rendezvous set.**

indicated by '$\dot{r}$' should remain parallel to LOS. If the particular rule stays intact throughout the motion of chasing vehicle, the distance between chaser and target would decrease till zero. The parallel navigation law is expressed by (2) and (3).

$$\boldsymbol{r} \times \dot{\boldsymbol{r}} = 0 \tag{2}$$

$$\boldsymbol{r} \cdot \dot{\boldsymbol{r}} < 0 \tag{3}$$

Eq (2) ensures that $\boldsymbol{r}$ and $\dot{\boldsymbol{r}}$ remains collinear while (3) ensures that $C$ is not receding from the target. By solving (2) and (3) in parametric form would result in (4).

$$\dot{\boldsymbol{r}} = -a\boldsymbol{r} \tag{4}$$

In (4), $a$ is a positive real number. The instantaneous relative velocity can now be written in the form of chasing vehicle and shadow target velocities which is given in (5).

$$\dot{\boldsymbol{r}} = \boldsymbol{v}_t - \boldsymbol{v}_c \tag{5}$$

By substituting (4) into (5), the resulting expression is given in (6).

$$\boldsymbol{v}_c = \boldsymbol{v}_t + a\boldsymbol{r} \tag{6}$$

The primary objective of trajectory planner is to obtain optimal chaser velocity command as per parallel navigation law for upcoming instant command. The value of $\boldsymbol{r}$ is obtained through proximity sensors installed on a vehicle. By substituting vector $\boldsymbol{r}$ in (6) would result in locus for the chasing vehicle velocity vectors $\boldsymbol{v}_c$ that lie in semi-line parameterized by $a$. This semi-line is known as Rendezvous line. The endpoints of velocity vectors $\boldsymbol{v}_c$ and $\boldsymbol{v}_t$ in indicate the positions of chaser and shadow target after a unit time. If chasing vehicle consistently follows velocity command that lies on RL, the direction of LOS remains constant which guarantees positional matching of chasing vehicle and shadow target. To find the value of $a$ given that velocity matching is realized, we assume that the acceleration of chaser in a given direction is indicated by $A$. The simultaneous reduction of velocity and position difference in the direction of LOS could be written in the form of (7).

$$\begin{cases} \dot{r}_{\max}^{rend} - A t_r = 0, \\ r - \dot{r}_{\max}^{rend} t r + \frac{1}{2} A t_r^2 = 0 \end{cases} \tag{7}$$

Where $\dot{r}_{\max}^{rend}$ is the magnitude of the maximum allowable closing velocity, and $t_r$ is the remaining *time-to-intercept* from the current instant. The maximum instantaneous allowable closing velocity is obtained by solving (7) which is given in (8).

$$\dot{r}_{\max}^{rend} = \sqrt{2rA} \tag{8}$$

The maximum *closing velocity*, as imposed by the frequency of velocity command generation by the trajectory planner for a fast asymptotic interception, is given in (9).

$$\dot{r}_{\max}^{cr} = {r}/{n.\Delta t} \tag{9}$$

The value of $n$ in (9) is determined empirically. The final value of permissible closing velocity component for velocity command is obtained by considering (8) and (9) which results in (10).

$$\boldsymbol{v}^{rel}_{\max} = \min \langle \dot{\boldsymbol{r}}^{rend}_{\max}, \dot{\boldsymbol{r}}^{cr}_{\max} \rangle \tag{10}$$

The endpoints of all velocity command vectors on $RL$ that have a closing velocity component smaller than $\boldsymbol{v}^{rel}_{\max}$ constitute a line segment extending from $\boldsymbol{v}_c = \boldsymbol{v}_t$ to $\boldsymbol{v}_c = \boldsymbol{v}_{c,\max}\left(= \boldsymbol{v}_t + \boldsymbol{v}^{rel}_{\max}\left(\boldsymbol{r}/{\|\boldsymbol{r}\|}\right)\right)$ which is given as red line in Fig 2. This set of points is referred to herein as the Rendezvous Set (RS). The velocity $\boldsymbol{v}^{rel}_{\max}$ may not be achieved by chasing vehicle within time interval $\Delta t$ due to constraints of user comfort and vehicle dynamics. Therefore, we need to determine a feasible region which includes a set of velocities that can be achieved by chasing vehicle within $\Delta t$ considering kinematic, dynamic, and passenger comfort constraints. This region is determined by imposing a limitation on lateral acceleration of chasing vehicle. We assume that maximum value for lateral acceleration of chaser vehicle is defined by (11).

$$a_{Y\max} = \frac{v^2_p}{K\mathrm{h}} 2\sin^2 \vartheta \left(1 + \frac{\cos \vartheta}{\sqrt{K^2 - \sin^2 \vartheta}}\right) \tag{11}$$

Where $a_{Y\max}$ is maximum lateral acceleration, $K = {v_p}/{v_s}$, h is the width of the lane, and $\vartheta$ is the maximum angle turning angle of chaser vehicle.

We assume that $\delta$ is current heading angle of chasing vehicle and by taking into consideration kinematic and dynamic vehicle constraints alongside user comfort constraints, feasible velocity region (FVR) is realized which is given in Fig 2. The velocity selected by vehicle for an interval $\Delta t$ is the component of RS that lies within FVR. The maximum velocity with FVR is $\boldsymbol{v}_c(t_i + \Delta t)$ as given in Fig 2. If chaser vehicle obeys the velocity commands having maximum velocity within FVR and velocity is also a component of RS then time-efficient interception can be realized.

**Modified Rendezvous Guidance algorithm.** The RG algorithm may not achieve predictability when it comes to robots but as the vehicle movements on highways is predictable so we can increase the velocity of 'C' to reduce lane changing and overtaking time. However, increase in velocity of 'C' is limited by lateral acceleration and user comfort constraints for the generation of trajectory command. The predictability provides us with an advantage to define a velocity line (VL). VL originates from a starting point of RL and makes an angle of $\vartheta$ with x-axis. Therefore, utilization of VL instead of RL for velocity command of upcoming instant ensures enhanced efficient lane changing and overtaking.

**Overtaking maneuver decisions for multiple scenarios.** As we have mentioned earlier that under the description of RG technique, shadow target will guide 'C' through all phases of overtaking. Therefore, a marker target ('M') which corresponds to shadow target is utilized for guiding 'C'. The location of 'M' depends upon the instant of lane changing and location of 'OB'. Initially, vision module gathers position and velocities of vehicles lies within the specific range. The information of vehicle velocity and position enables algorithm to compute RS and eventually yields a closing velocity component $\boldsymbol{v}^{rel}_{\max}$. The desired velocity to be realized by 'C' in upcoming instant via modified RG method is represented by $\boldsymbol{v}_{RG}$. To obtain desired value of acceleration, the acceleration command ($\boldsymbol{a}_{RG}$) is fed to 'C' for next instance. $\boldsymbol{v}^{rel}_{\max}$ must lie within the FVR and if that's not the case then optimal velocity is selected from RS for the

upcoming instance. A new feasible velocity region (NFVR) is formed by the intersection of VL with FVR which contains velocities to be generated for upcoming instant commands. $\mathbf{v}_1$ and $\mathbf{v}_2$ are two intersection points where VL intersects FVR as given in Fig 3 and selection of any point within NFVR ensures a rendezvous with '$M$'. The selection of maximum velocity from NFVR which is $\mathbf{v}_1$ will enable '$C$' to get nearest to '$M$'. Therefore, for upcoming instant, the required velocity for '$C$' could be written in the form of (12).

$$\mathbf{v}_{RG} = \mathbf{v}_c(t_i + \Delta t) = \mathbf{v}_1 \tag{12}$$

The corresponding lateral acceleration command could be written in terms of (13).

$$\mathbf{a}_{RG} = \frac{\mathbf{v}_{RG} - \mathbf{v}_c(t_i)}{\Delta t} \tag{13}$$

**Overtaking scenario 1: Absence of vehicle in overtaking lane.** The absence of any vehicle in overtaking lane makes overtaking maneuver quite simple for '$C$'. As the distance between '$C$' and '$L$' gets to 2.5s, the proposed algorithm will check if there is any vehicle in overtaking lane or not? If an overtaking lane is free of any obstacle vehicle, the vehicles will continue to move forward till the distance between them gets 2s. The RG algorithm will come to effect at a distance of 2s by the creation of imaginary marker targets 'M1', 'M2', and 'M3' for guiding '$C$' throughout the overtaking maneuver. During overtaking, the velocity of '$C$' remains constant within each particular phase. The pictorial representation of overtaking scenario 1 is given in Fig 4. Upper lane is overtaking lane while lower lane is travelling lane.

**Overtaking scenario 2: Presence of vehicle in overtaking lane.** The presence of any vehicle in overtaking lane makes overtaking maneuver a bit tricky for '$C$'. As the distance between '$C$' and '$L$' gets to 2.5s, the proposed algorithm will check if there is any vehicle in overtaking lane or not? If overtaking lane has any obstacle vehicle '$OB$', then '$C$' will wait for '$OB$' to overtake '$L$' first so that it can perform overtaking maneuver with safety. In this scenario, '$M_1$' is created 2s behind '$L$' and velocity of '$M_1$' is equal to '$L$'. Once overtaking lane is cleared of '$OB$', '$C$' can start overtaking maneuvering. However, in this scenario, velocities of '$M_2$' and '$M_3$' can either be set to initial velocity of '$C$' which is $\mathbf{v}_s = \mathbf{v}_t$ or initial velocity of '$OB$' which is $\mathbf{v}_s = \mathbf{v}_B$.

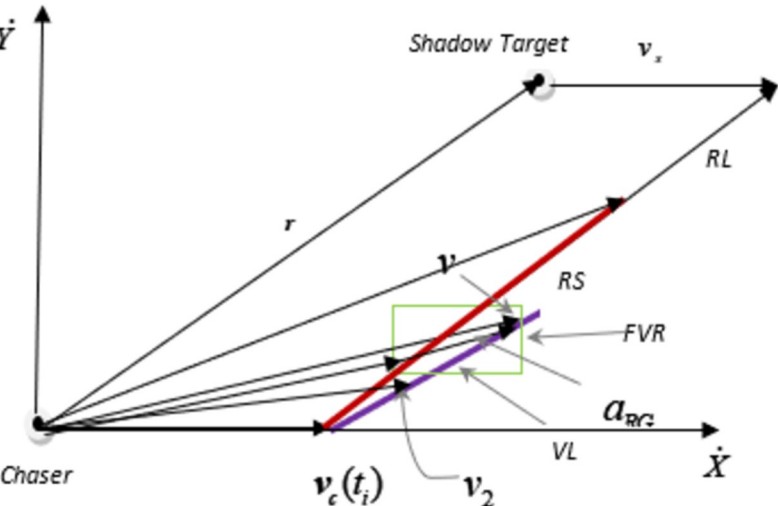

**Fig 3. Generation of command for chasing vehicle.**

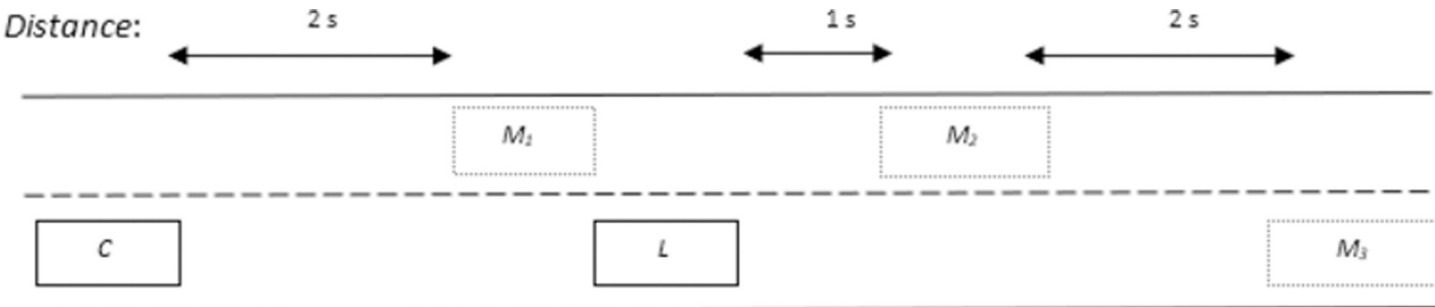

**Fig 4. Position of marker targets for scenario 1.**

The selection between the two velocities will be done by utilization of velocity with less magnitude. The pictorial representation of overtaking scenario 2 is given in Fig 5.

**Overtaking scenario 3: Quitting overtaking maneuver.** The overtaking scenario may get complicated when 'C' starts overtaking maneuver and suddenly 'L' accelerates to increase its velocity such that its velocity gets greater than 'C' ($v_l > v_c$). Therefore, 'C' cannot overtake 'L' in this scenario so algorithm will decide to abort the overtaking process and allows 'C' to travel with the same velocity in overtaking lane till the distance between them gets 3s. Now, the algorithm creates a marker target 'M' in driving lane to allow 'C' to get back into the driving lane. The position of 'M' is stationary and a pictorial representation of overtaking scenario 3 is given in Fig 6.

**Overtaking scenario 4: Overtaking maneuvering with multiple vehicles.** The overtaking scenario gets complex with the involvement of overtaking multiple vehicles and presence of 'OB' in overtaking lane. By assuming a scenario in which there are two vehicles '$L_1$' and '$L_2$' in driving lane that 'C' needs to takeover and '$L_2$' is ahead of '$L_1$'. The velocities of '$L_1$' and '$L_2$' are $v_{L1}$ and $v_{L2}$ respectively. 'OB' is also moving in overtaking lane so due to safety considerations; RG algorithm will not permit 'C' to perform overtaking straightaway. In addition, the distance between '$L_1$' and '$L_2$' is lower enough so that 'C' cannot return to driving lane before overtaking both vehicles. Therefore, under such circumstances, a new location of marker is added 1s ahead of '$L_1$' which is given as '$M_3$' in Fig 7. Therefore, the given set of markers in Fig 7 will allow 'C' to overtake '$L_1$' and '$L_2$' alongside avoiding collision with 'OB' in overtaking lane.

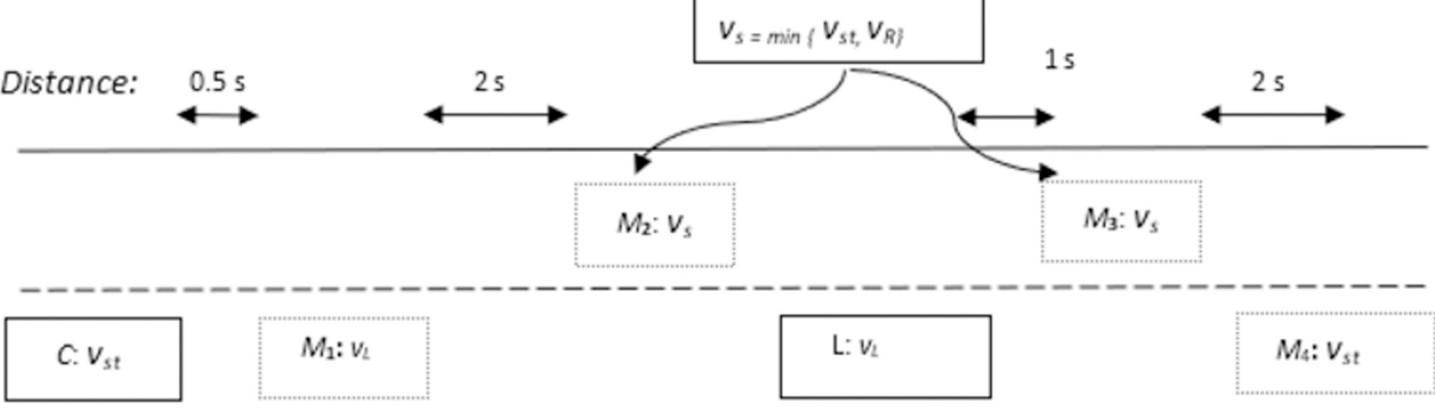

**Fig 5. Position of marker targets for scenario 2.**

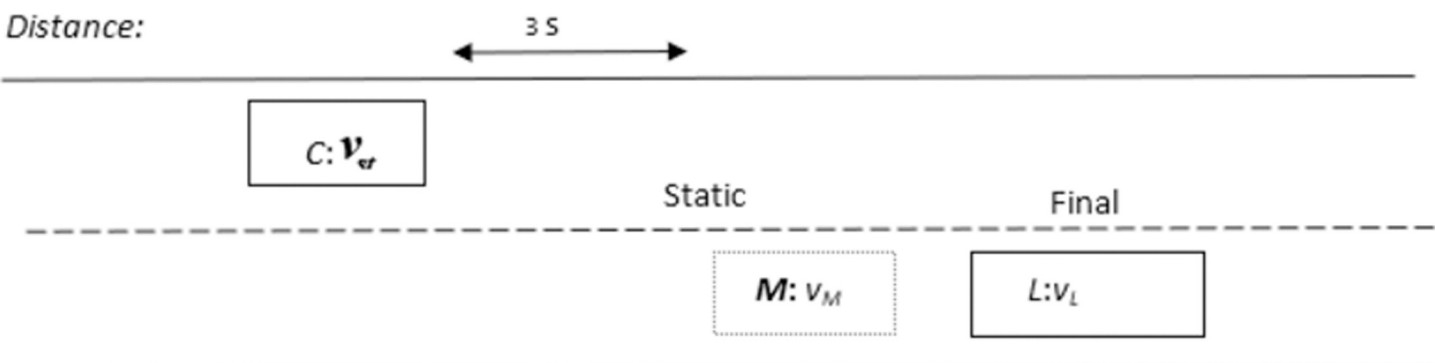

**Fig 6. Position of marker targets for scenario 3.**

## Simulation results

To demonstrate the effectiveness of the proposed algorithm, several simulations were conducted to confirm the accuracy of their response to the location of leading and obstacle vehicles. Various combinations of '*C*', '*L*', and '*OB*' parameters have been tested to check the efficacy of proposed algorithm. The obtained results indicate that proposed algorithm is effective in guiding chasing vehicle to perform overtaking maneuver comfortably without any sort of accidents. The comparative analysis of proposed algorithm and RG algorithm is also performed to highlight the benefits of modified RG algorithm over the conventional RG algorithm.

**Modified RG algorithm results.** The simulation results for all four overtaking scenarios with modified form of RG algorithm are discussed separately.

*Overtaking scenario 1*. In this scenario, chasing vehicle needs to overtake leading vehicle and there is no blocking vehicle obstructing the path of chasing vehicle in overtaking lane. The preliminary velocity of '*C*' is taken as 30 m/s and that of '*L*' as 20 m/s. '*L*' is accelerating to achieve a velocity of 22 m/s. The velocity and path of '*C*' are given in Fig 8. The lateral and axial velocities of chasing vehicle are given in Fig 9. The obtained values of time taken for

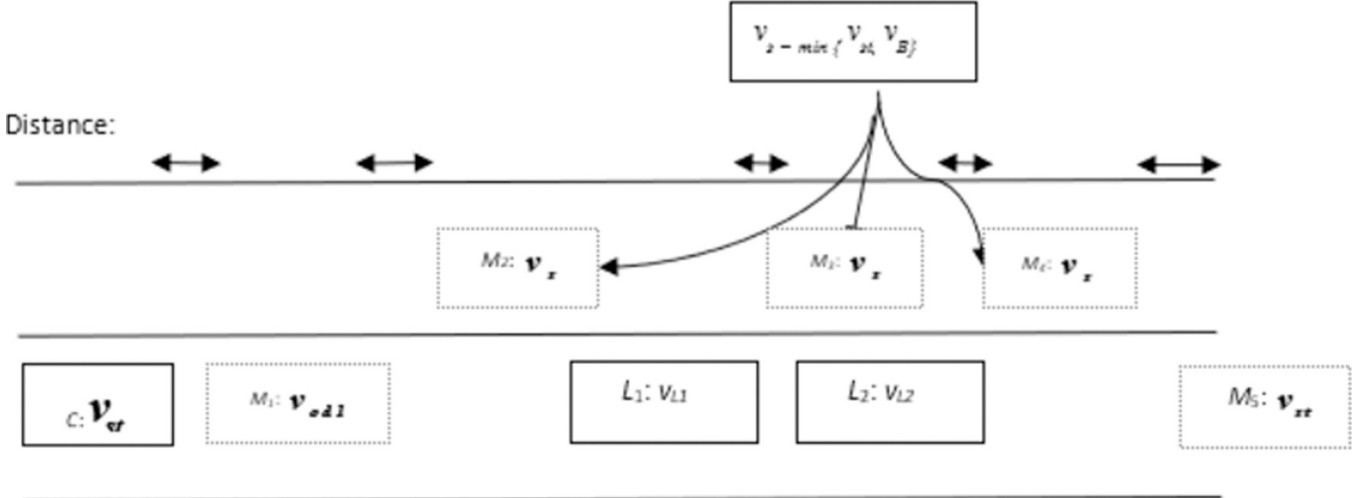

**Fig 7. Position of marker targets for scenario 4.**

overtaking, maximum velocity achieved, maximum lateral acceleration, and maximum axial acceleration are listed in Table 1.

*Overtaking scenario 2.* In this scenario, chasing vehicle needs to overtake leading vehicle and blocking vehicle is also obstructing the path of chasing vehicle in overtaking lane. The velocities of 'L' and 'OB' are varying sinusoidally and had an initial velocity of 20 m/s and 27.5 m/s respectively. The velocity and path of 'C' are given in Fig 10. The lateral and axial velocities of chasing vehicle are given in Fig 11. The driving lane and passing lane obstacle velocities for chasing vehicle are given in Fig 12. The obtained values of time taken for overtaking, maximum velocity achieved, maximum lateral acceleration, and maximum axial acceleration are listed in Table 2.

*Overtaking scenario 3.* In this scenario, once chasing vehicle enters overtaking lane and starts to overtake leading vehicle, the leading vehicle accelerates, and its velocity gets greater than the chasing vehicle. Therefore, chasing vehicle needs to quit overtaking maneuver and get back in the driving lane. As 'C' starts to overtake 'L', 'L' starts accelerating to a velocity of 34 m/s which leads 'C' to abort overtaking maneuver. The velocity and path of 'C' are given in Fig 13.

*Overtaking scenario 4.* In this scenario, chasing vehicle needs to overtake multiple (two) leading vehicles in driving lane and blocking vehicle is also obstructing the path of chasing vehicle in overtaking lane. The velocities of '$L_1$' and '$L_2$' are 20 m/s and the preliminary velocity of 'C' is 30 m/s. The velocity and path of 'C' are given in Fig 14.

**Comparative analysis of modified and conventional RG algorithm.** The comparative analysis required that simulations which were performed for modified RG algorithm should be repetitively performed for the conventional RG method. All four scenarios are utilized for comparison of both methods. Tabular representation of the parameter values of the total time taken for vehicle to perform overtaking maneuver and distance travelled during overtaking maneuvering are done for each scenario for readers to grasp which method performs better.

*Overtaking scenario 1.* In this scenario, chasing vehicle needs to overtake leading vehicle and there is no blocking vehicle obstructing the path of chasing vehicle in overtaking lane. 'L' is deaccelerating to drop its velocity from 22 m/s to 18 m/s. The parameter values for both methods are listed in Table 3. In this case, time taken for overtaking maneuver in modified RG method is less which means that chasing vehicle has to cover less distance to complete overtaking maneuvering.

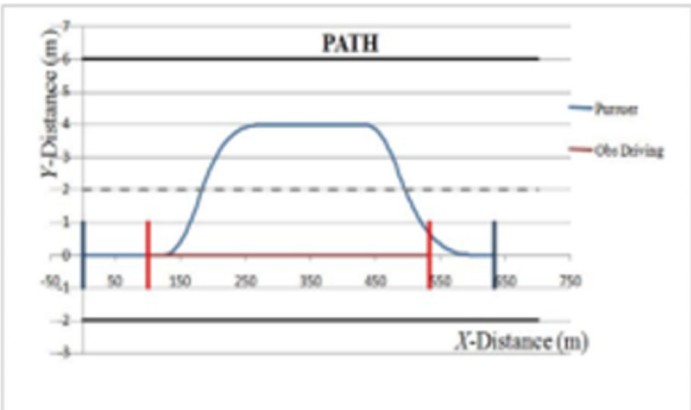

**Fig 8. Profiling–velocity and path for C in scenario 1.**

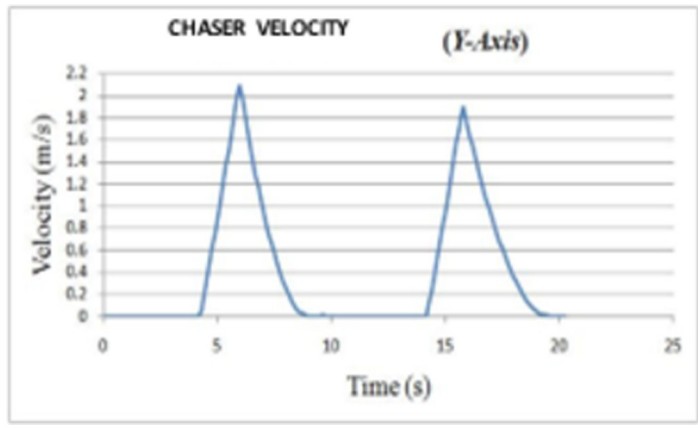 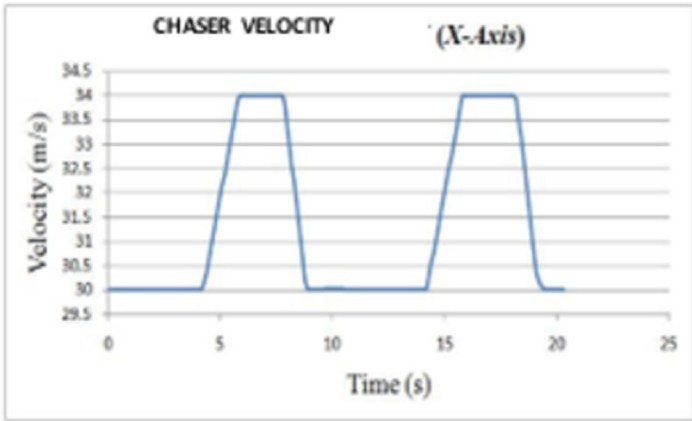

**Fig 9. Lateral and axial velocities of C in scenario 1.**

*Overtaking scenario 2*. In this scenario, chasing vehicle needs to overtake leading vehicle and blocking vehicle is obstructing the path of chasing vehicle in overtaking lane. '*L*' is moving with constant velocity and '*C*' is accelerating which increases its velocity from 25 m/s to 30 m/ s. The parameter values for both methods are listed in Table 4. In this case, time taken for over- taking maneuver in modified RG method is less which means that chasing vehicle has to cover less distance to complete overtaking maneuvering.

*Overtaking scenario 3*. In this scenario, chasing vehicle wants to overtake leading vehicle but speed of leading vehicle is more than chasing vehicle so chasing vehicle comes down to lower lane without overtaking. Initially starting speed of chasing vehicle is 20 m/s while speed of vehicle to overtaken is 25 m/s. during overtaking speed of vehicle to be overtaken increases to 30 m/s. The parameter values for both methods are listed in Table 5. In this case, time taken for returning to travelling in modified RG method is less which means that chasing vehicle has to cover less distance to complete overtaking maneuvering.

*Overtaking scenario 4*. In this scenario, chasing vehicle needs to overtake two leading vehicle and there is also blocking vehicle obstructing the path of chasing vehicle in overtaking lane. Initial velocity of chasing vehicle is 25m/s while vehicles to overtaken are having velocity of 20m/s. The parameter values for both methods are listed in Table 6. In this case, time taken for overtaking maneuver in modified RG method is less which means that chasing vehicle has to cover less distance to complete overtaking maneuvering.

**Comparative analysis of RG algorithm with conventional off-line overtaking method.** The comparison of the proposed on-line methodology was also carried with an off-line method presented in [38]. Since the technique proposed cannot cope with variations in obsta- cle velocity, for this comparison, both vehicles are moving with constant velocities. Simula- tions are performed for all four scenarios in this comparison. The results of the simulations for the comparison are presented via table. The table shows the time taken and the distance trav- elled using both methodologies.

**Table 1. Summarized results for scenario 1.**

| S.No. | Parameters | Values |
|---|---|---|
| 1 | Total time taken for overtaking (s) | 16.3 |
| 2 | Maximum velocity achieved (m/s) | 34 |
| 3 | Maximum lateral acceleration (m/s$^2$) | 1.04 |
| 4 | Maximum axial acceleration (m/s$^2$) | 2.5 |

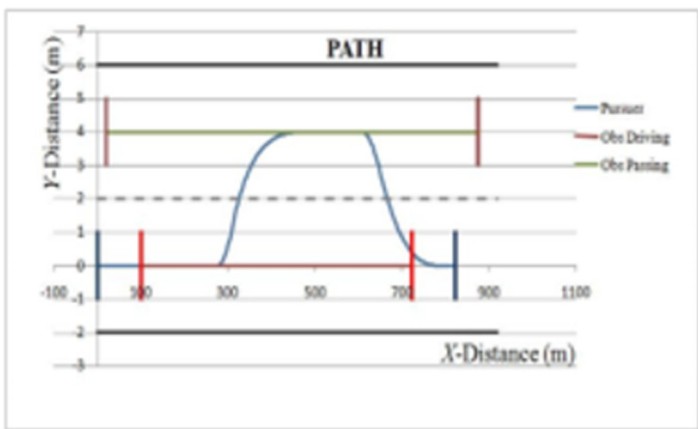
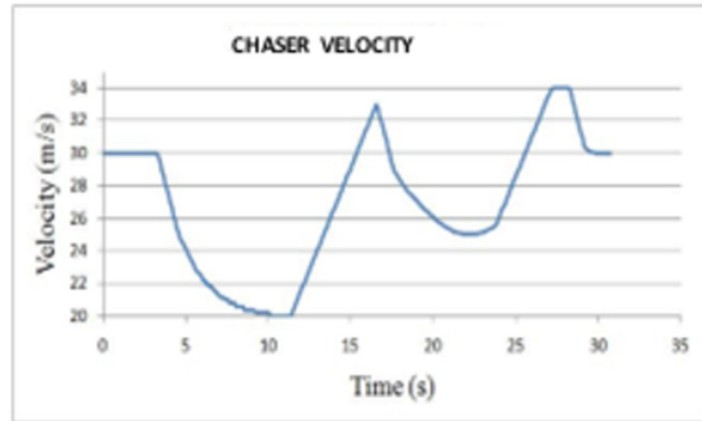

**Fig 10. Profiling–velocity and path for C in scenario 2.**

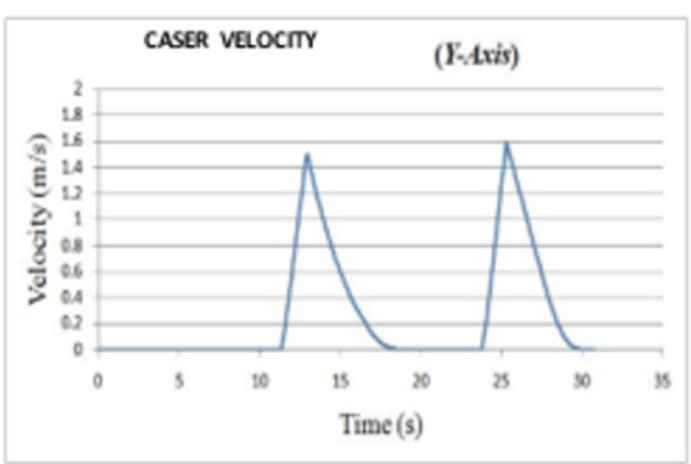
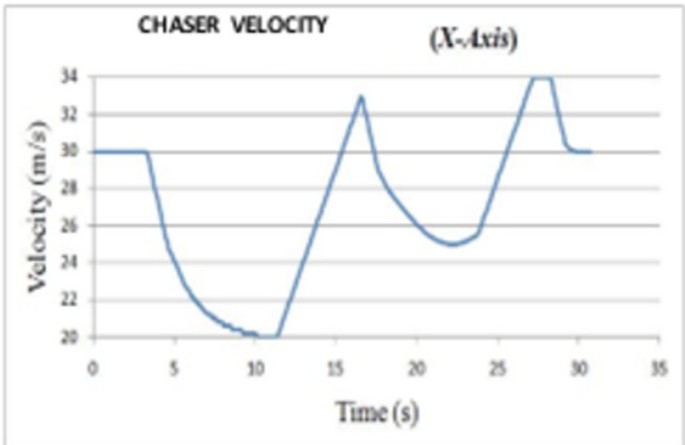

**Fig 11. Lateral and axial velocities of C in scenario 2.**

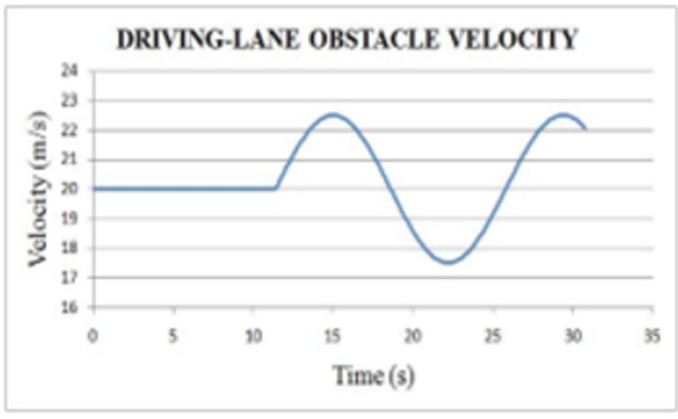
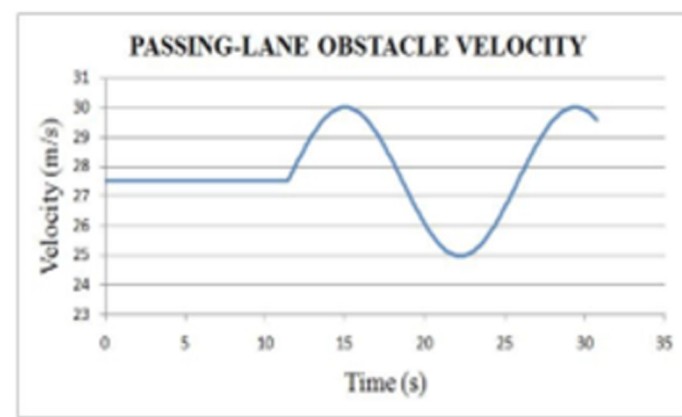

**Fig 12. Driving lane and passing lane obstacle velocities.**

**Table 2. Summarized results for scenario 2.**

| S.No. | Parameters | Values |
|---|---|---|
| 1 | Total time taken for overtaking (s) | 26.8 |
| 2 | Maximum velocity achieved (m/s) | 34 |
| 3 | Maximum lateral acceleration (m/s$^2$) | 0.88 |
| 4 | Maximum axial acceleration (m/s$^2$) | 2.35 |

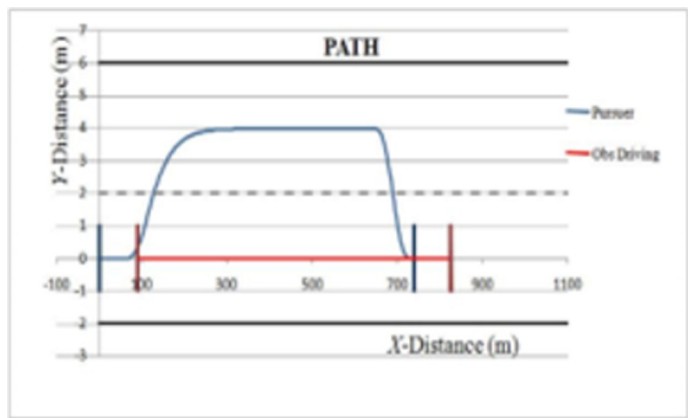
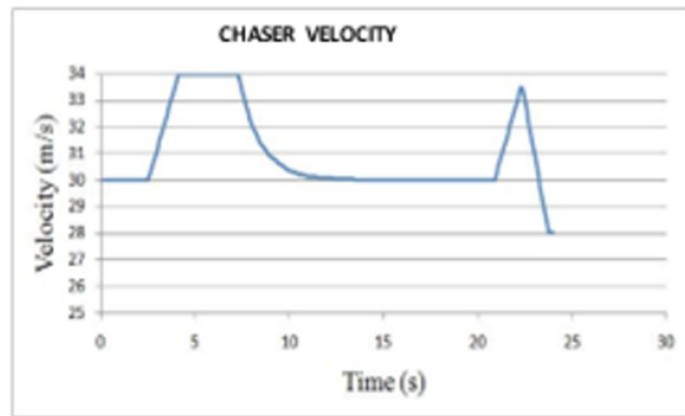

**Fig 13. Profiling–velocity and path for C in scenario 3.**

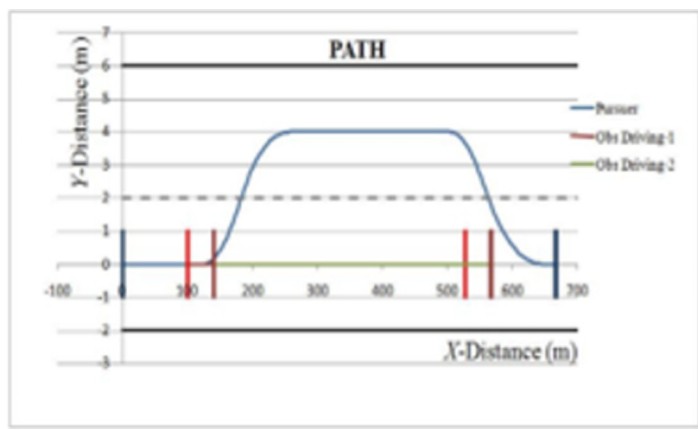
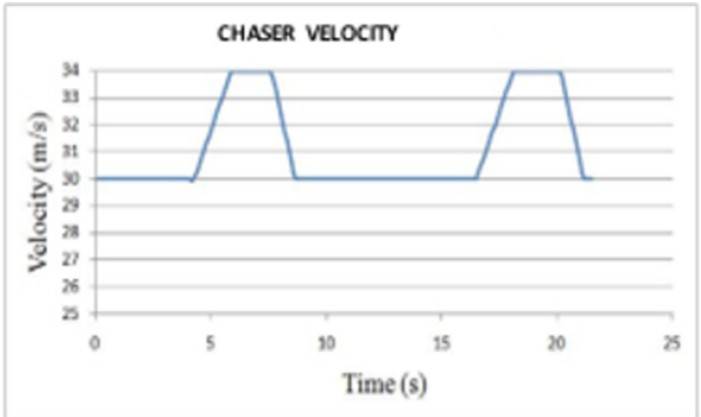

**Fig 14. Profiling–velocity and path for C in scenario 4.**

**Table 3. Comparison of time and distance in scenario 1.**

| Parameters | Modified RG Method | Conventional RG Method |
|---|---|---|
| Total time (s) | 11.5 | 12.7 |
| Distance travelled (m) | 368 | 390 |

**Table 4. Comparison of time and distance in scenario 2.**

| Parameters | Modified RG Method | Conventional RG Method |
|---|---|---|
| Total time (s) | 28.4 | 28.8 |
| Distance travelled (m) | 728 | 736 |

**Table 5. Comparison of time and distance in scenario 3.**

| Parameters | Modified RG Method | Conventional RG Method |
|---|---|---|
| Total time (s) | 24 | 26 |
| Distance travelled (m) | 730 | 740 |

*Scenario 1*. There is no obstacle vehicle present which could delay/restrict the overtaking manoeuvre. In this scenario, vehicle to be overtaken, is moving with a constant velocity of 20 m/s and the velocity of C is 30 m/s. Table 7 shows basic overtaking parameters of both methods for scenario 1.

*Scenario 2*. There is another vehicle present in overtaking lane which could delay/restrict the overtaking manoeuvre. In this scenario, both vehicles are moving with constant velocities of 20 m/s and 25 m/s, respectively. The starting velocity of C is taken as 30 m/s. Table 8 shows basic overtaking parameters of both methods for scenario 2.

*Scenario 3*. Chasing vehicle is having slow speed as compared to vehicle to be overtaking, so overtaking does not happen. In this scenario, chasing vehicle and vehicle to be overtaken are moving with initial velocities of 25 m/s and 20 m/s, respectively. Velocity of vehicle to be overtaken increases to 30m/s, so the chasing vehicle comes back to travelling lane. Table 9 shows basic overtaking parameters of both methods for scenario 2.

*Scenario 4*. Chasing vehicle is to overtake vehicle with double lanes and in the presence of obstructing vehicle in overtaking lane. Table 10 shows basic overtaking parameters of both methods for scenario 2.

## Experimental setup

Numerous experiments were conducted by having different numbers of chasing and leading vehicles. With the help of these experimental trials, the comparative analysis of modified and conventional RG method is performed. The experimental results clearly indicate a 10% decrease in overtaking maneuver period by modified RG method. The experimental trials are conducted using robots and it illustrates almost similar behavior as its simulations. The image of workspace is captured and processed to gather information of all objects within workspace. The information gathering from workspace is performed with the help of image acquisition and processing module of software. The extracted information is then used for trajectory planning which is used for the calculation of command to be sent to chasing vehicle.

The specifications of hardware used in the experiment are given in Table 11. The software used for the experiment comprises of three primary modules which include image acquisition and processing, trajectory planning, and communication modules. At first stage, analog CCD camera captures image of workspace and second stage involves extraction of positional information by vision algorithm. Third stage is about forwarding extracted information to trajectory planner for computation of acceleration command (real-time) for chasing vehicle.

**Table 6. Comparison of time and distance in scenario 3.**

| Parameters | Modified RG Method | Conventional RG Method |
|---|---|---|
| Total time (s) | 22 | 25 |
| Distance travelled (m) | 690 | 725 |

**Table 7. Basic overtaking parameters for scenario 1 –a comparison.**

|                        | Modified RG Method | Off-Line Method |
|------------------------|--------------------|-----------------|
| Total time (s)         | 13.5               | 15.5            |
| Distance travelled (m) | 430                | 466             |

**Table 8. Basic overtaking parameters for scenario 2 –a comparison.**

|                        | Modified RG Method | Off-Line Method |
|------------------------|--------------------|-----------------|
| Total time (s)         | 34.2               | 39.5            |
| Distance travelled (m) | 844                | 966             |

**Table 9. Basic overtaking parameters for scenario 3 –a comparison.**

|                        | Modified RG Method | Off-Line Method |
|------------------------|--------------------|-----------------|
| Total time (s)         | 23                 | 27              |
| Distance travelled (m) | 720                | 770             |

**Table 10. Basic overtaking parameters for scenario 4 –a comparison.**

|                        | Modified RG Method | Off-Line Method |
|------------------------|--------------------|-----------------|
| Total time (s)         | 22                 | 25              |
| Distance travelled (m) | 690                | 725             |

**Table 11. Experimental hardware specifications.**

| Component | Characteristics |
|-----------|-----------------|
| Pursuer and Obstacle Vehicles | Miabot PRO BT v2 Differential-Drive mobile Robots with Bluetooth Communication |
| CCD Camera | Resolution: 640 × 480 pixels |
|  | Lens Focal Length: 6 mm |
|  | Vertical Distance from Floor: 3000 mm |
| Floor Workspace | 2740 × 1500 mm |

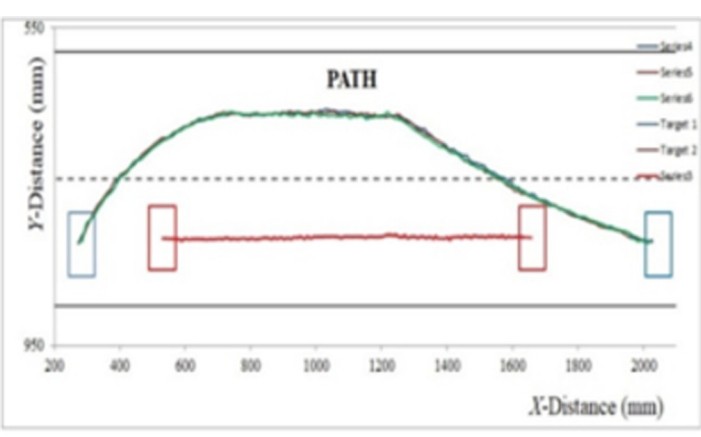 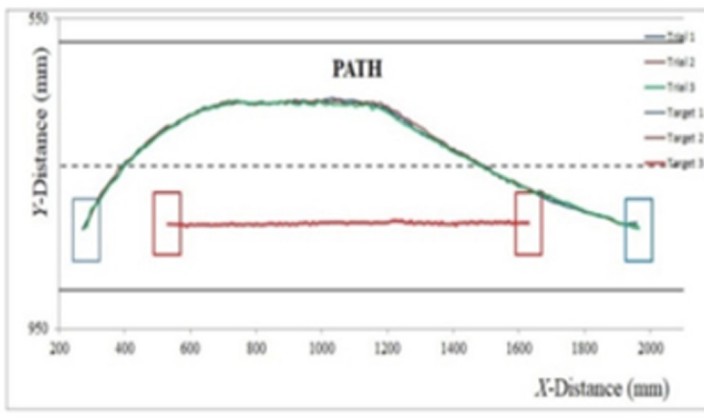

**Fig 15. Marker target position for situation 1.**

### First experiment

In this experiment, the overtaking lane is free of blocking vehicle. The width of each lane is 160 mm, initial velocity of '$C$' is 8 mm/s and initial velocity of '$L$' is 6 mm/s. The image of marker targets '$M$' are given in Fig 15 for each phase during overtaking maneuvering. The comparison between experimental and simulated results is demonstrated with the help of simulation results in Fig 16 and experimental results in Fig 17. Three experimental trials are conducted with similar parameters under identical conditions. The results demonstrate that modified RG method has less time period for completing overtaking maneuvering. The conventional RG method yields overtaking distance of 2200 mm and modified RG method results in a distance of 2000 mm. Thus, a decrease of 9.8% is witnessed using modified RG method.

### Second experiment

In this experiment, the overtaking lane has a presence of blocking vehicle. The width of each lane is 160 mm, initial velocity of '$C$' is 8 mm/s, initial velocity of '$L$' is 6 mm/s, and initial velocity of '$OB$' is 8 mm/s. The results demonstrate that modified RG method has less time period for completing overtaking maneuvering. The conventional RG method yields overtaking distance of 2227 mm and modified RG method results in a distance of 2075 mm. Thus, a decrease of 9% is witnessed using modified RG method.

The takeaways from experimental results are as follows:

- In a noisy environment, RG method performs overtaking maneuvering free of accidents which illustrates the robustness of approach.

- The experimental results bolster the simulation results by demonstrating similar paths with and without the presence of blocking vehicle in overtaking lane. In addition, it indicates that the proposed modified approach performs better in comparison with the conventional RG method.

### Conclusion

This article presents robust and time-optimal guidance-based algorithm for trajectory planning. The proposed approach not only adapts to environmental changes rather reacts to them in an appropriate manner that is particular to a given situation. The proposed modified RG algorithm is simulated for four different scenarios which involve an absence

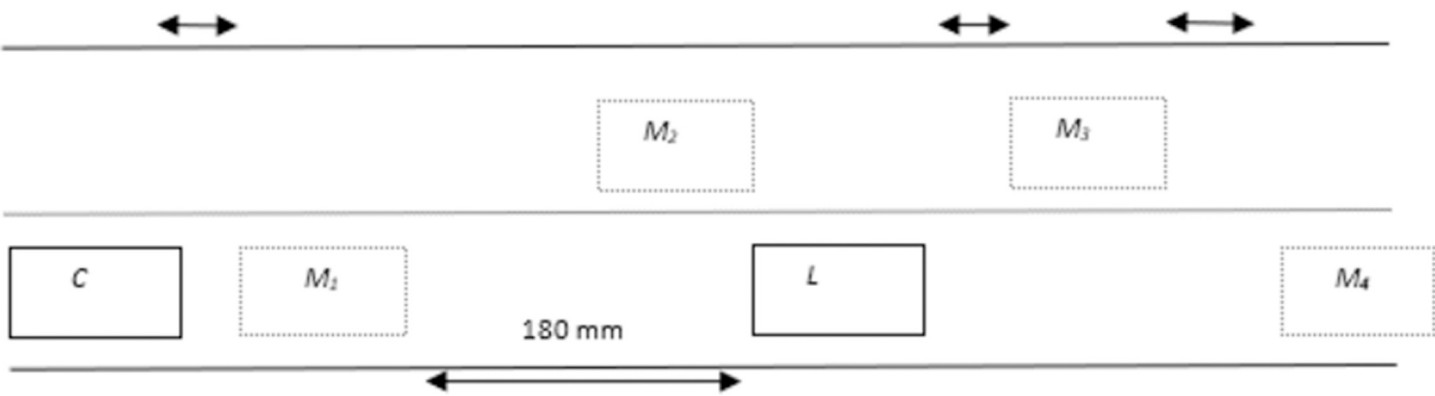

**Fig 16.** Experiment 1- simulated path (a) Original RG method (b) Modified RG method.

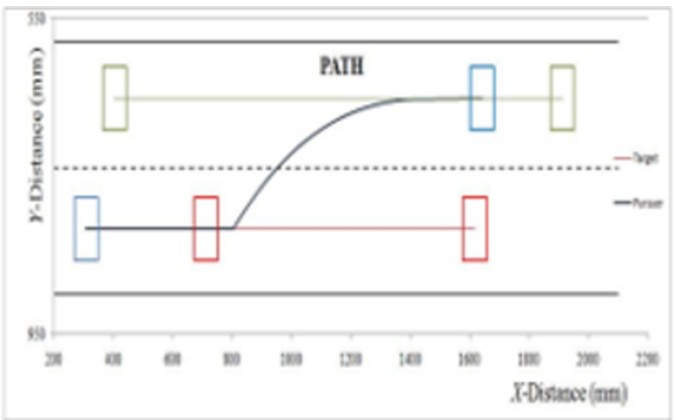 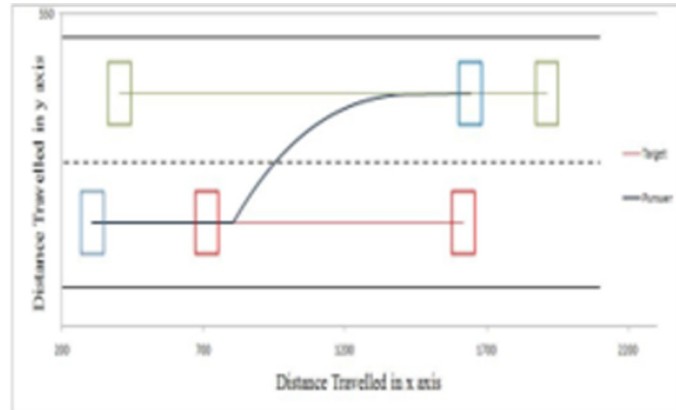

**Fig 17.** Experiment 1- experimental path (a) Original RG method (b) Modified RG method.

of '*OB*' in overtaking lane, presence of '*OB*' in overtaking lane, single leading vehicle, and multiple leading vehicles in driving lane. The simulation results in all four scenarios reflect accident-free overtaking maneuvering completion. In addition, comparative analysis for simulation results of conventional and modified RG method is performed using two of the four scenarios discussed. The modified RG approach enables overtaking maneuvering time to decrease by 10%. After establishing the significance of modified RG method over the conventional RG method, it is required to check if the simulated results of modified method are comparable to its experimental results or not? Therefore, for similar two situations, the experimental results are compared with simulated results and experimental results support simulated results as they are almost identical. The presented modified RG method ensures accident-free overtaking in all scenarios which makes it better than off-line solutions suggested by previous works. However, future work may concentrate on comparative analysis of experimental and simulation results of both approaches with complex scenarios having multiple blocking and leading vehicles in overtaking and driving lane respectively.

## Supporting information

**S1 File.**
(RAR)

**S2 File.**
(ZIP)

**S3 File.**
(ZIP)

**S4 File.**
(RAR)

## Author Contributions

**Conceptualization:** Usman Ghumman, Ihsan Ullah Khalil.

**Investigation:** Usman Ghumman.

**Methodology:** Usman Ghumman.

**Project administration:** Mohsin Islam Tiwana.

**Resources:** Mohsin Islam Tiwana.

**Supervision:** Hamid Jabbar.

**Writing – original draft:** Ihsan Ullah Khalil.

**Writing – review & editing:** Ihsan Ullah Khalil, Faraz Kunwar.

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
