## [Decision Letter · Decision Letter 0]

13 Sep 2021

PONE-D-21-28089A Novel Approach of Overtaking Maneuvering using Modified RG MethodPLOS ONE

Dear Dr. Usman Ghumman,

Thank you for submitting your manuscript to PLOS ONE. After careful consideration, we feel that it has merit but does not fully meet PLOS ONE’s publication criteria as it currently stands. Therefore, we invite you to submit a revised version of the manuscript that addresses the points raised during the review process.

We look forward to receiving your revised manuscript.

Kind regards,

Jing Zhao, Ph.D.

Academic Editor

PLOS ONE

Journal Requirements:

3. We note that Figure 28 in your submission contain copyrighted images. All PLOS content is published under the Creative Commons Attribution License (CC BY 4.0), which means that the manuscript, images, and Supporting Information files will be freely available online, and any third party is permitted to access, download, copy, distribute, and use these materials in any way, even commercially, with proper attribution. For more information, see our copyright guidelines: http://journals.plos.org/plosone/s/licenses-and-copyright.

a. You may seek permission from the original copyright holder of Figure 28 to publish the content specifically under the CC BY 4.0 license. 

Reviewers' comments:

Reviewer's Responses to Questions

**Comments to the Author**

1. Is the manuscript technically sound, and do the data support the conclusions?

Reviewer #1: Partly

Reviewer #2: Yes

2. Has the statistical analysis been performed appropriately and rigorously? 

Reviewer #1: I Don't Know

Reviewer #2: Yes

3. Have the authors made all data underlying the findings in their manuscript fully available?

Reviewer #1: Yes

Reviewer #2: Yes

4. Is the manuscript presented in an intelligible fashion and written in standard English?

Reviewer #1: Yes

Reviewer #2: Yes

5. Review Comments to the Author

Reviewer #1: 1、The overtaking manoeuvre is a 2-dimensional movement. The literature review is poor. No relevant studies are introduced. At least, the following two references should be cited:

(1) Zhao, J., Knoop, V.L., Wang, M., 2020. Two-dimensional vehicular movement modelling at intersections based on optimal control. Transportation Research Part B: Methodological 138, 1-22.

(2) Bichiou, Y., Rakha, H.A., 2018. Developing an optimal intersection control system for automated connected vehicles. IEEE Trans. Intell. Transp. Syst. 20, 1908–1916.

2、RG technique is originally introduced for spaceships rendezvous missions with space stations and asteroids. Why is RG technology used in overtaking in this paper? What are the advantages of this method over traditional overtaking methods?

3、The scope of application of this method should be explained. leading vehicle may change speed, lane, etc. is this method applicable? In addition, the influence of lane changing behavior of rear vehicles on overtaking cannot be ignored.

4、The improved RG technology is better than that before the improvement. This conclusion is easy to draw. But will the improved RG model be better than the traditional classical overtaking models? The improved method should be compared with the classical overtaking models.

Reviewer #2: This paper focuses on the overtaking maneuvering for trajectory planning and obstacle avoidance. The topic is interesting, but the treatment of the methodology is weak. The authors should make great efforts to make this paper published.

The layout of this paper should obey the format.

The figures are unclear and ambiguous for readers. And it is undesirable to include so many figures in this paper.

Many relative scientific papers are missing in the Introduction section.

The contributions of this paper is unclear.

The paper contains many typos and errors of units.

The Rendezvous Guidance technique should be expressed clearly in section 2.2.

How does this algorithm distinguish the overtaking scenarios? Sometimes two or more scenarios may occur at the same time. For instance, there are multiple vehicles but the subject vehicle achieves the unsuccessful overtaking. Will this algorithm handle this situation?

In figure 24, it seems the two vehicles collide with each other, which is unreasonable.

6. PLOS authors have the option to publish the peer review history of their article (what does this mean?). If published, this will include your full peer review and any attached files.

Reviewer #1: No

Reviewer #2: No

---

## [Author Response · Author response to Decision Letter 0]

9 Oct 2021

All data sets of experiments, scenarios and simulation codes along with simulation videos are uploaded upon request of editor as a confirmation of data avalibility. All the concerns and recommendations of editors and reviewers are addressed upto the best effort

---

## [Decision Letter · Decision Letter 1]

1 Nov 2021

PONE-D-21-28089R1A Novel Approach of Overtaking Maneuvering using Modified RG MethodPLOS ONE

Dear Dr. Usman Ghumman,

Thank you for submitting your manuscript to PLOS ONE. After careful consideration, we feel that it has merit but does not fully meet PLOS ONE’s publication criteria as it currently stands. Therefore, we invite you to submit a revised version of the manuscript that addresses the points raised during the review process.

We look forward to receiving your revised manuscript.

Kind regards,

Jing Zhao, Ph.D.

Academic Editor

PLOS ONE

Journal Requirements:

Additional Editor Comments (if provided):

Reviewers' comments:

Reviewer's Responses to Questions

**Comments to the Author**

1. If the authors have adequately addressed your comments raised in a previous round of review and you feel that this manuscript is now acceptable for publication, you may indicate that here to bypass the “Comments to the Author” section, enter your conflict of interest statement in the “Confidential to Editor” section, and submit your "Accept" recommendation.

Reviewer #1: All comments have been addressed

Reviewer #2: (No Response)

2. Is the manuscript technically sound, and do the data support the conclusions?

Reviewer #1: Yes

Reviewer #2: Yes

3. Has the statistical analysis been performed appropriately and rigorously? 

Reviewer #1: Yes

Reviewer #2: No

4. Have the authors made all data underlying the findings in their manuscript fully available?

Reviewer #1: Yes

Reviewer #2: (No Response)

5. Is the manuscript presented in an intelligible fashion and written in standard English?

Reviewer #1: Yes

Reviewer #2: (No Response)

6. Review Comments to the Author

Reviewer #1: A novel modified form of RG technique is proposed and its comparative analysis with conventional RG technique is also conducted to validate the effectiveness of the proposed approach. The proposed technique yields better results as it allows 10% less time for overtaking. Modified RG technique is also compared with conventional offline technique. The proposed technique is technically sound, and the data support the conclusions.

Reviewer #2: Although there are so many figures in the revised manuscript (which however are not shown), the presented figures convey little information and the explanation of the figures are quite poor. For instance, where is the overtaking lane and OB in Figure 7 to Figure 10?

Many figures in the compressed file are still unclear. In addition, I still hold the idea that too many figures without sufficient information may distract readers, so summarizing and combining these figures are necessary. I hesitantly appreciate the authors' action of concern 3.

I did not see 6 scenarios in this paper. There are actually 4 overtaking scenarios. Be careful when using "case", "scenario" and "situation".

As to Figure 11, the y-distance path subfigure fails to show the time in second, so relationship between the speeds and the lateral position is unclear. The same holds for the other similar figures.

The comparison between the modified and conventional RG algorithms under scenario 3 and 4 should also be included.

More comparison cases with the off-line overtaking method are encouraged, since scenario 1 is quite simple.

Why the indicator values of the modified RG method in Table 3, 5 and 6 are different? The simulation settings when comparing with the conventional RG and the off-line overtaking method are supposed to be identical.

There are still many grammar and typo errors throughout the paper. Please check.

7. PLOS authors have the option to publish the peer review history of their article (what does this mean?). If published, this will include your full peer review and any attached files.

Reviewer #1: No

Reviewer #2: No

---

## [Author Response · Author response to Decision Letter 1]

4 Nov 2021

Minor revison and concerns of reviewer#2 are adressed.

---

## [Decision Letter · Decision Letter 2]

10 Nov 2021

A Novel Approach of Overtaking Maneuvering using Modified RG Method

PONE-D-21-28089R2

Dear Dr. Usman Ghumman,

We’re pleased to inform you that your manuscript has been judged scientifically suitable for publication and will be formally accepted for publication once it meets all outstanding technical requirements.

Kind regards,

Jing Zhao, Ph.D.

Academic Editor

PLOS ONE

Additional Editor Comments (optional):

Reviewers' comments:

Reviewer's Responses to Questions

**Comments to the Author**

1. If the authors have adequately addressed your comments raised in a previous round of review and you feel that this manuscript is now acceptable for publication, you may indicate that here to bypass the “Comments to the Author” section, enter your conflict of interest statement in the “Confidential to Editor” section, and submit your "Accept" recommendation.

Reviewer #1: All comments have been addressed

Reviewer #2: (No Response)

2. Is the manuscript technically sound, and do the data support the conclusions?

Reviewer #1: Yes

Reviewer #2: (No Response)

3. Has the statistical analysis been performed appropriately and rigorously? 

Reviewer #1: Yes

Reviewer #2: (No Response)

4. Have the authors made all data underlying the findings in their manuscript fully available?

Reviewer #1: Yes

Reviewer #2: (No Response)

5. Is the manuscript presented in an intelligible fashion and written in standard English?

Reviewer #1: Yes

Reviewer #2: (No Response)

6. Review Comments to the Author

Reviewer #1: A novel modified form of RG technique is proposed and its comparative analysis with conventional RG technique is also conducted to validate the effectiveness of the proposed approach. The proposed technique yields better results as it allows 10% less time for overtaking.

Reviewer #2: (No Response)

7. PLOS authors have the option to publish the peer review history of their article (what does this mean?). If published, this will include your full peer review and any attached files.

Reviewer #1: No

Reviewer #2: No

---

## [Editor Report · Acceptance letter]

10 Dec 2021

PONE-D-21-28089R2 

 A novel approach of overtaking maneuvering using modified RG method 

Dear Dr. Ghumman:

I'm pleased to inform you that your manuscript has been deemed suitable for publication in PLOS ONE. Congratulations! Your manuscript is now with our production department. 

Kind regards, 

on behalf of

Dr. Jing Zhao 

Academic Editor

PLOS ONE